# ADAPT-TO-LEARN: POLICY TRANSFER IN REINFORCEMENT LEARNING

## ABSTRACT

Efficient and robust policy transfer remains a key challenge in reinforcement learning. Policy transfer through warm initialization, imitation, or interacting over a large set of agents with randomized instances, have been commonly applied to solve a variety of Reinforcement Learning (RL) tasks. However, this is far from how behavior transfer happens in the biological world: Humans and animals are able to quickly adapt the learned behaviors between similar tasks and learn new skills when presented with new situations. Here we seek to answer the question: Will learning to combine adaptation reward with environmental reward lead to a more efficient transfer of policies between domains? We introduce a principled mechanism that can **"Adapt-to-Learn"**, that is adapt the source policy to learn to solve a target task with significant transition differences and uncertainties. We show through theory and experiments that our method leads to a significantly reduced sample complexity of transferring the policies between the tasks.

## 1 INTRODUCTION

Lack of principled mechanisms to quickly and efficiently transfer policies learned between domains has become the major bottleneck in Reinforcement Learning (RL). This inability to transfer or adapt policies is one major reason why RL has still not proliferated physical application like robotics. Since RL agents cannot quickly transfer policies, the agent is forced to learn every task from scratch, which is both time and sample expensive. Warm-start, a method in which weights from one neural network are transferred to another, has been reasonably successful for supervised learning. However, this method can often lead to mixed and even negative results in RL (Joshi & Chowdhary, 2018; Taylor & Stone, 2009).

Our main contribution is an algorithm to transfer policies between tasks with significant differences in state transitions via a policy adaptation mechanism. Unlike the majority of existing work in transfer learning for RL, our approach does not merely use the transferred policy to warm start (initialize the parameter of the target network with learned source network) policy learning in the target domain. Neither does it rely on a multitude of simulations across randomly generated source domains. Instead, we combine supervised reference trajectory tracking and unsupervised reinforcement learning to adapt the source policy to the target domain directly. We show through theory and experiments that our method enjoys significantly reduced sample complexity in solving the task.

Adapt-to-Learn is inspired by the fact that combined adaptation of behaviors and learning through experience is a primary mechanism of learning in biological creatures (Krakauer & Mazzoni, 2011; Fryling et al., 2011).Inspired by this ability of biological creatures, we seek to answer the question: Will learning to combine intrinsic adaptation reward with environment reward lead to more efficient transfer of policies between domains? Imitation Learning (IL) (Duan et al., 2017; Zhu et al., 2018) seems to play a crucial part in biological learning, and as such has been widely studied in RL. However, the key is, when presented with a new situation, animals do not just imitate, but quickly adapt existing behaviors, and improve them through further experience. In particular, an animal learning to walk on a different terrain does not just imitate its existing gait, but adapts it to the new environment.The theory behind such adaptation in reference tracking control problems has been typically restricted to deterministic dynamical systems with well-defined reference trajectories (Åström & Wittenmark, 2013; Chowdhary et al., 2013). This ability to adapt and incorporate further learning through optimization on the environment reward is one key difference between our method and ex-

isting imitation learning and Guided Policy Search (GPS) methods (Levine & Koltun, 2013). Unlike IL and GPS, our method transfers policies between task with significant differences in the transition models. Moreover, by mixing environment reward with intrinsic adaptation rewards, we ensure that the agent quickly adapts and also learns to acquire skills beyond what the source policy can teach. We posit that the presented method can be the foundation of a broader class of RL algorithms that can choose seamlessly between learning through RL to supervised adaptive imitation. Our empirical results show that approach is capable of robustly transferring policies between tasks, even in the presence of nonlinear and time-varying differences in the dynamic model of the systems. In particular, we show that it suffices to execute adapted greedy policies to ensure $\epsilon-$optimal behavior in the target domain.

**Related work:** D-RL has recently enabled agents to learn policies for complex robotic tasks in simulation (Peng et al., 2016; 2017b; Liu & Hodgins, 2017; Heess et al., 2017). However, D-RL has been plagued by the curse of sample complexity. Therefore, the capabilities demonstrated in the simulated environment are hard to replicate in the real world. This learning inefficiency of RL has led to significant work in the field of TL (Taylor & Stone, 2009). A significant body of literature on transfer in RL is focused on initialized RL in the target domain using learned source policy; known as jump-start/warm-start methods (Taylor et al., 2005; Ammar et al., 2012; 2015). Some examples of these transfer architectures include transfer between similar tasks (Banerjee & Stone, 2007), transfer from human demonstrations (Peters & Schaal, 2006) and transfer from simulation to real (Peng et al., 2017a; Ross et al., 2011; Yan et al., 2017). Efforts have also been made in exploring accelerated learning directly on real robots, through Guided Policy Search (GPS) (Levine et al., 2015) and parallelizing the training across multiple agents using meta-learning (Levine et al., 2016; Nagabandi et al., 2018; Zhu et al., 2018). Sim-to-Real transfers have been widely adopted in the recent works and can be viewed as a subset of same domain transfer problems. Daftry et al. (Daftry et al., 2016) demonstrated the policy transfer for control of aerial vehicles across different vehicle models and environments. Christiano et al. (Christiano et al., 2016) transferred policies from simulation to real using an inverse dynamics model estimated interacting with the real robot. Through learning over an adversarial loss, the agents are trained to achieve robust policies across various environments (Wulfmeier et al., 2017). However, these and other reported architectures do not *necessarily* lead to improved sample efficiency, handle relatively minor changes in the transition model, and are even known to cause negative transfer. In contrast, our approach directly adapts the source policies to target with significant transition model difference while interacting with the environment. It enjoys empirically and rigorously proven sample efficiency guarantees of order $\mathcal{O}(n\mathcal{H})$, depending polynomially on the horizon length "$\mathcal{H}$".

## 2 PRELIMINARIES

Consider a finite horizon MDP defined as a tuple $\mathcal{M} = (\mathcal{S}, \mathcal{A}, \mathcal{P}, \mathcal{R}, \rho_0, \gamma)$, where $\mathcal{S}$ denote set of continuous states; $\mathcal{A}$ is a set of continuous bounded actions, $\mathcal{P} : \mathcal{S} \times \mathcal{A} \times \mathcal{S} \to \mathbb{R}_+$ is state transition probability distribution of reaching $s'$ upon taking action $a$ in $s$, $\rho_0 : \mathcal{S} \to \mathbb{R}_+$ is the distribution over initial states $s_0$ and $\mathcal{R} : \mathcal{S} \times \mathcal{A} \to \mathbb{R}_+$ is deterministic reward function and $\mathcal{H}$ be the finite horizon of the problem.

Let $\pi(a|s) : \mathcal{S} \times \mathcal{A} \to [0, 1]$ be stochastic policy over continuous state and action space. The action from policy is a draw from this distribution $a_i \sim \pi(a_i|s_i)$. The agent's goal is to find a policy $\pi^\star$ which maximize the total return. The total return starting from states $s_0 \sim \rho_0$ under a policy $\pi$ is

$$\eta_\pi(s_0) = \mathbb{E}_{s_0, a_0, \dots} \left( \sum_{t=0}^{\mathcal{H}} r(s_t, a_t) \right) \tag{1}$$

where, $s_0 \sim \rho_0$, $a_t \sim \pi(a_t|s_t)$ and $s_{t+1} \sim p(s_{t+1}|s_t, a_t)$.

We formalize the underlying problem of policy transfer by considering a source and target MDP as follows, $\mathcal{M}^S = (\mathcal{S}, \mathcal{A}, \mathcal{P}, \mathcal{R}, \rho_0, \gamma)^S$, $\mathcal{M}^T = (\mathcal{S}, \mathcal{A}, \mathcal{P}, \mathcal{R}, \rho_0, \gamma)^T$, each with its own state, action space and transition model respectively. We will mainly focus on the problem of same domain transfer in this paper, where the state and action space are analogous $\mathcal{S}^{(S)} = \mathcal{S}^{(T)} = \mathcal{S} \in \mathbb{R}^m$ and $\mathcal{A}^{(T)} = \mathcal{A}^{(S)} = \mathcal{A} \in \mathbb{R}^k$, but the source and target state transition models differ significantly due to unmodeled dynamics or external environment interactions. Furthermore, an example of how the method can extend to cross-domain transfer using manifold alignment (Joshi & Chowdhary,

2018; Wang & Mahadevan, 2009) is in the supplementary. Let $\pi^*$ be a parameterized optimal stochastic policy for source MDP $\mathcal{M}^S$. The policy $\pi^*$ can be obtained using any available RL methods (Sutton et al., 2000; Schulman et al., 2015; 2017). In this work, we have used the Proximal Policy Optimization(PPO) (Schulman et al., 2017) algorithm to generate the source optimal policy. We will use the following definition of the state value function $V^\pi$ defined under any policy $\pi$

$$V^\pi(s_t) = \mathbb{E}_{a_t, s_{t+1}, a_{t+1}, \ldots} \left( \sum_{i=0}^{\mathcal{H}} r(s_{i+t}, a_{i+t}) \right)$$

The associated optimal value function fo source MDP $\mathcal{M}^S$ is $V^{\pi^*}$ for all $s \in \mathcal{S}$. Let $\mathcal{T} : \mathbb{R}^n \to \mathbb{R}^n$ be the Bellman update operator defined as

$$(\mathcal{T}f)(s) = \max_{a \in \mathcal{A}} \left[ r(s, a) + \gamma \mathop{\mathbb{E}}_{s' \sim P(s,a)} f(s') \right] \tag{2}$$

Optimal value function $V^{\pi^*}$ satisfies the Bellman Equation-(2) such that $V^{\pi^*}(s) = \mathcal{T}V^{\pi^*}(s), \forall\, s$.

# 3 ALGORITHMIC CONTRIBUTION: TRANSFER THROUGH ADAPTATION

## 3.1 ADAPTATION AS A MECHANISM FOR POLICY TRANSFER IN RL

In this paper, we approach the problem of transfer learning for RL through adaptation of previously learned policies to related tasks. Our approach to adaptation is by enabling the agent to learn the best mixture of imitation and learning from environment reward.Our method differs from RL transfer methods that rely on jump-starts or direct imitation (Zhu et al., 2018; Duan et al., 2017) in a key way: We do not aim to emulate the source optimal policy itself in the target domain, but the source transitions under optimal policy. To make this point, we demonstrate in our empirical results that the source policy, when used directly, does not produce sensible behaviors of the agent in the target domain when transition models are significantly different.On the other hand, our method uses the source transitions projected onto the target task as reference exploration trajectories, helping to adapt and optimize the source policy to the target domain efficiently.

In canonical reinforcement learning, the goal is to learn the optimal policy $\pi^*$ which maximizes the future cumulative reward when the reward function and the transition function of the MDP are unknown. In statistical RL, this leads to the (in)famous explore-exploit tradeoff.Yet, this learn-through-experience approach has demonstrated significant potential to enable general solutions to control of complex physical systems for which first principles-based models are not easily obtainable. However, one of the key challenges in RL has been the high sample complexity of obtaining a reasonable policy. In other words, RL, and specifically Deep RL, require an absurdly huge amount of interactions with the environment (experience) to find reasonable policies. This high learning inefficiency is in apparent contrast with the efficient learning demonstrated by humans and many biological creatures. For examples, octopuses are known to learn to solve complex problems such as opening jars and squeezing through puzzles with just a few trials. This ability to quickly transfer and adapt learned policies between related tasks is likely a key to general intelligence that seems to be missing from RL. Our goal in the rest of this paper is to show that an algorithm that can judiciously combine adaptation to changes and learning new skills is capable of avoiding brute force random exploration to a large extent and be significantly less sample expensive.

## 3.2 ADAPT-TO-LEARN: POLICY TRANSFER

We begin by mathematically describing adaptation for policy transfer in RL and state all the necessary assumptions in Section 3.2.1. We then develop the Adapt-to-Learn algorithm in Section 3.2.2.

### 3.2.1 ADAPTATION

The proposed Adapt-to-Learn (ATL) algorithm objective is to accumulate higher total returns through maximizing the likelihood of the target policies $\pi_\theta$, which emulate the optimal source transition behavior in the target MDP. We achieve this objective by minimizing the KL divergence between point-wise local trajectories realized using the target policy and source optimal policy in the target

domain (Refer Figure-3 in Appendix-C). This adaptation objective can be formalized as minimizing the KL-divergence between source and target transition trajectories:

$$\eta_{KL}(\pi_\theta, \pi^*) = \mathbb{E}_{s_t, a_t \sim \tau} \left( p_{\pi_\theta}(\tau) \log \left( \frac{p_{\pi_\theta}(\tau)}{q_{\pi^*}(\tau)} \right) \right), \tag{3}$$

where $\tau$ is the trajectory in the target domain under the policy $\pi_\theta$ defined as

$$\tau = (s_0, a_0, s_1, a_1, \dots),$$

and probability of the trajectory $p_{\pi_\theta}(\tau)$ under policy $\pi_\theta$ and target transition $p^T(.|s_t, a_t)$ can be written as

$$p_{\pi_\theta}(\tau) = \rho(s_0) \prod_{t=0}^{\mathcal{H}} \pi_\theta(a_t|s_t) p^T(s_{t+1}|s_t, a_t). \tag{4}$$

Similarly $q_{\pi^*}(\tau)$ can be defined as probability of trajectory deviations at every state $s_t \in \tau$, when the source optimal policy $\pi^*$ is used in place of target policy $\pi_\theta$, and the states evolve according to the source transitions model $p^S(.|s_t, a_t)$ (Refer Figure-3 in Appendix-C):

$$q_{\pi^*}(\tau) = \rho(s_0) \prod_{t=0}^{\mathcal{H}} \pi^*(a_t'|s_t) p^S(s_{t+1}'|s_t, a_t'). \tag{5}$$

***Assumption-1***: We assume that optimal source policy is available, such that the source policy variance can be assumed to be zero. That is, the action probabilities $\pi^*(a_t'|s_t) = 1, \forall s_t$. This is a reasonable assumption, since optimal source policy is available it can be treated as deterministic policy and actions can be chosen greedily. We need this assumption only for deriving the expression for intrinsic reward and theoretical analysis. In the empirical evaluation of the algorithm, we treat $\pi^*(.)$ as a stochastic policy.

The transition probabilities in the KL divergence in (3) term are treated as transition likelihoods and the transitioned state $s_{t+1}$ as a random variable. Since $s_{t+1}'$ is the optimal state reached starting in $s_t$ under source transition model using $\pi^*(a_t'|s_t)$, we try to emulate this behavior in the target domain and hence evaluate the transition likelihoods of the target trajectory at $\{s_i'\}_{i=1}^{\mathcal{H}}$. Using the definition of the probabilities of the trajectories under $\pi_\theta$ and $\pi^*$ Equation-(4) & (5) the log term in the KL divergence of the trajectory (3) is simplified as follows

$$\log \left( \frac{p_{\pi_\theta}(\tau)}{q_{\pi^*}(\tau)} \right) = \log \left( \frac{\rho(s_0)\pi(a_0|s_0)p^T(s_1'|s_0, a_0)\pi(a_1|s_1)p^T(s_2'|s_1, a_1)\dots}{\rho(s_0)\pi^*(a_0'|s_0)p^S(s_1'|s_0, a_0')\pi^*(a_1'|s_1)p^S(s_2'|s_1, a_1')\dots} \right). \tag{6}$$

Using (6) and with assumption-1 i.e. $\pi^*(a_t'|s_t) = 1 \forall s_t$ the KL term can simplified as follows

$$\eta_{KL}(\pi_\theta, \pi^*) = \mathbb{E}_{s_t, a_t \sim \tau} \left( p_{\pi_\theta}(\tau) \sum_{t=0}^{\mathcal{H}} \log \left( \frac{\pi_\theta(a_t|s_t)p^T(s_{t+1}'|s_t, a_t)}{p^S(s_{t+1}'|s_t, a_t')} \right) \right)$$

$$\eta_{KL}(\pi_\theta, \pi^*) = \mathbb{E}_{s_t, a_t \sim \tau} \left( p_{\pi_\theta}(\tau) \sum_{t=0}^{\mathcal{H}} \zeta_t \right) \tag{7}$$

where $\zeta_t = \log \left( \frac{\pi_\theta(a_t|s_t)p^T(s_{t+1}'|s_t, a_t)}{p^S(s_{t+1}'|s_t, a_t')} \right)$.

If calculating the above expectation is feasible, it is possible to minimize the KL divergence and move the policy parameters in the direction of emulating source transition behavior in the target domain. However, this is not generally the case since the true expectation is intractable. Therefore a common practice is to use an empirical estimate of the expectation to do approximate planning. We minimize the trajectory KL divergence by handling the term $\zeta_t$ as the intrinsic adaptation reward, which captures the local deviation in trajectory in the form of the shaped reward function.

***Assumption-2***: We do not assume to know the true transition distribution for source and the target, but only have access to simulator model of the source. However, we assume both source and target transition models follow a Gaussian distribution centered at the next propagated state and fixed variance "$\sigma$".

Assumption-2 is not very restrictive since we empirically show that for any deterministic model, a bootstrapped Gaussian transition assumption is sufficient for ATL agent to learn the task. Using the above assumption, we can approximate the KL term as

$$\zeta_t = \log\left(\pi_\theta(a_t|s_t)e^{-(s_{t+1}-s'_{t+1})^2/2\sigma^2}\right). \tag{8}$$

The individual terms in the expectation $\zeta_t$ represent the distance between two transition likelihoods of landing in the next state $s'_{t+1}$ starting in $s_t$ and under actions $a_t, a'_t$. The target agent is encouraged to take actions that lead to states which are close in expectation to a reference state provided by an optimal baseline policy operating on the source model. By doing so, we are providing a possible direction of search for the higher environmental rewards "$r_t$".

We can, therefore, solve the following optimization problem to generate adaptive policy updates:

$$\pi_\theta^{*T} = \arg\min_{\pi_\theta\in\Pi}\left(\eta_{KL}\right). \tag{9}$$

The distance between two transition models $\zeta_t$ is used as an intrinsic reward to calculate the total trajectory intrinsic return, and further, any policy update algorithm can be used (Sutton et al., 2000; Schulman et al., 2017; 2015) to update the policy in direction optimizing this objective. However, it is to be noted that though we use policy gradient kind of update for adaptive policy, the optimization is more akin to supervised learning. For every state-action pair, the KL-distance is the true metric to be minimized, and unlike RL, we do not engage in optimal value search in the adaptation part of learning. Hence the algorithm is more sample efficient compared to any RL policy search methods.

### 3.2.2 Adaptation and Learning

We achieve Adaption and Learning simultaneously by augmenting environment reward $r_t$ with intrinsic reward $\zeta_t$. By doing so, we achieve transferred policies which try to both optimize optimal reference tracking and also maximize the cumulative future environmental reward to acquire skills beyond what source can teach. This trade-off between learning by exploration and learning by adaptation can be realized as follows:

$$\eta_{KL} = \mathbb{E}_{s_t,a_t}\left(p_{\pi_\theta}(\tau)\sum_{t=0}^{\mathcal{H}}((1-\beta)r_t - \beta\zeta_t)\right). \tag{10}$$

Where the term $\beta$ is the mixing coefficient. We make a heuristic choice for an appropriate $\beta$ to start with and annealed over episodes for optimal mixing of adaptation and learning. For consistency of the reward mixing, the rewards $r_t, \zeta_t$ are normalized to form the total reward $r'_t = (1-\beta)r_t + \beta\zeta_t$.

---

**Algorithm 1** Adaptive-to-Learn Policy Transfer in RL

**Require:** $\pi^*(.), p^S$                   ▷ Inputs: Source Policy, source simulator
1: Initialize $s_0^T \in \rho_0$.           ▷ Draw initial state from the given distribution in target task
2: **for** $i = 1 \leq K$ **do**
3:     **while** $s_i \neq terminal$ **do**
4:         $a'_i \sim \pi^*(s_i)$            ▷ Generate the optimal action using Source policy
5:         $a_i \sim \pi_\theta(s_i)$               ▷ Generate the action using the $\hat{\pi}_\theta$
6:         $s_{i+1} \sim p^T(s_i, a_i)$       ▷ Propagate the target task model at state $s_i$ and action $a_i$
7:         $s'_{i+1} \sim p^S(s_i, a'_i)$      ▷ Propagate the source task model at state $s_i$ and action $a'_i$
8:         $\zeta_t = \pi_\theta(a_i|s_i)e^{-(s_{i+1}-s'_{i+1})^2/2\sigma^2}$    ▷ Calculate the KL divergence intrinsic reward term
9:         $Z_i = (\{s_i, r_i, \zeta_i, a_i, a'_i\})$     ▷ Incrementally store the trajectory for policy update
10:    $P_{Z^n}(\eta_{KL}) = \frac{1}{n}\sum_{i=1}^{n}\left(\sum_{t=0}^{\mathcal{H}}\nabla_\theta\log\pi_\theta(a_{i,t}|s_{i,t})\sum_{t=0}^{\infty}r'_{i,t}\right)$   ▷ Empirical loss to optimize
11:    $\hat{\pi}_\theta = arg\max_{\pi\in\Pi} P_{Z^n}(\eta_{KL})$        ▷ Minimize the loss to obtain a adaptive policy

---

### 3.3 Sample-Based estimation of the gradient

The previous section proposed an optimization method to find the adaptive policy using KL-divergence as an intrinsic reward, enforcing the target transition model to mimic the source transitions. This section describes how this objective can be approximated using a Monte Carlo simulation. The adaptive policy update methods work by computing an estimator of the gradient of the

return and plugging it into a stochastic gradient ascent algorithm.

$$\hat{\pi}_\theta^{*T} = arg \max_{\pi_\theta \in \Pi} P_{Z^n}(\eta_{KL}) \tag{11}$$

$$\theta \leftarrow \theta + \alpha \hat{g}.$$

Where $\alpha$ is the learning rate and $\hat{g}$ is the empirical estimate of the gradient (Refer Appendix-A) of the total return $\eta_{KL}$. The gradient estimate over i.i.d data from the collected trajectories is computed as follows:

$$\hat{g} = P_{Z^n}(\nabla_\theta \eta_{KL}) = \hat{\mathbb{E}}_{s_t,a_t \sim \tau} \left( \sum_{t=0}^{\mathcal{H}} \nabla_\theta \log \pi_\theta(a_t|s_t) \sum_{t=0}^{\infty} r_t' \right),$$

$$P_{Z^n}(\nabla_\theta \eta_{KL}) = \frac{1}{n} \sum_{i=1}^{n} \left( \sum_{t=0}^{\mathcal{H}} \nabla_\theta \log \pi_\theta(a_{i,t}|s_{i,t}) \sum_{t=0}^{\infty} r_{i,t}' \right), \tag{12}$$

where $P_{Z^n}$ is empirical distribution over the data ($Z^n : \{s_i, a_i, a_i'\}_i^n$).

## 4    THEORETICAL BOUNDS ON SAMPLE COMPLEXITY

Although there is some empirical evidence that transfer can improve performance in subsequent reinforcement-learning tasks, there are not many theoretical guarantees. Since many of the existing transfer algorithms approach the problem of transfer as a method of providing good initialization to target task RL, we can expect the sample complexity of those algorithms to still be a function of the cardinality of state-action pairs $|N| = |\mathcal{S}| \times |\mathcal{A}|$. On the other hand, in supervised learning setting, the theoretical guarantees of most algorithm have no dependency on size (or dimensionality) of the input domain (which is analogous to $|N|$ in RL domains). Having formulated a policy transfer algorithm using labeled reference trajectories derived from optimal source policy in ATL, we construct supervised learning like PAC property of the proposed method. For the sample complexity analysis we consider only the adaptation part of learning i.e. $\beta = 1$ in Equation-(16). This is because in ATL, the adaptive learning is akin to supervised learning, since the source reference trajectories provide the target states given every $(s_t, a_t)$ pair.

Suppose we are given the learning problem specified with training set $Z^n = (Z1, \ldots Z_n)$ where each $Z_i = (\{s_i, a_i, a_i'\})_{i=0}^{\mathcal{H}}$ are independently drawn according to some distribution $P$. Given the data $Z^n$ we can compute the empirical return $P_{Z^n}(\eta_{KL})$ for every $\pi_\theta \in \Pi$, and we will show that the following holds:

$$\|P_{Z^n}(\eta_{KL}) - P(\eta_{KL})\| \leq \epsilon \tag{13}$$

with probability at least $1 - \delta$, for some very small $\delta$ s.t $0 \leq \delta \leq 1$. We can claim that the empirical return for all $\pi_\theta$ is a sufficiently accurate estimate of the true return function. Thus a reasonable learning strategy is to find a $\pi_\theta \in \Pi$ that would minimize empirical estimate of the objective (10).

**Theorem 4.1** *If the induced class of the policy $\pi_\theta$:$\mathcal{L}_\Pi$ has uniform convergence property in empirical mean; then the empirical risk minimization is PAC. s.t*

$$P^n(P(\eta_{KL,\hat{\pi}^*}) - P(\eta_{KL,\pi^*}) \geq \epsilon) \leq \delta \tag{14}$$

*and number of trajectory samples required can be lower bounded as*

$$n(\epsilon, \delta) \geq \frac{2\mathcal{H}^2 C^2}{\epsilon^2} \log\left(\frac{2|\Pi|}{\delta}\right). \tag{15}$$

For the proof of the above theorem refer the Appendix-B.

### 4.1    $\epsilon$-OPTIMALITY RESULT UNDER ADAPTIVE TRANSFER-LEARNING

Consider MDP $M^*$ and $\hat{M}$ which differ in their transition models. For the sake of analysis, let $M^*$ be the MDP with ideal transition model, such that target follows source transition $p^*$ precisely. Let $\hat{p}$ be the transition model achieved (tracked) using the estimated adapted policy learned over data interacting with the target model and the associated MDP be denoted as $\hat{M}$. We analyze the $\epsilon$-optimality of return under adapted source optimal policy through proposed policy transfer.

Note for the sake of analysis we are deviating from finite horizon assumption and using discout factor $\gamma$. To be consistent with previous section we can assume the horizon length $\mathcal{H} = \log_\gamma \left( \frac{\epsilon_v}{2V_{max}} \right)$, where $\epsilon_v$ is truncation error in total infinite return.

**Definition 4.2** *Given the value function $V^* = V^{\pi^*}$ and model $M_1$ and $M_2$, which only differ in the corresponding transition models $p_1$ and $p_2$. Lets define $\forall s, a \in \mathcal{S} \times \mathcal{A}$*

$$d_{M_1, M_2}^{V^*} = \sup_{s, a \in \mathcal{S} \times \mathcal{A}} \left| \mathbb{E}_{s' \sim P_1(s,a)} [V^*(s')] - \mathbb{E}_{s' \sim P_2(s,a)} [V^*(s')] \right|$$

**Lemma 4.3** *Given $M^*$, $\hat{M}$ and value function $V_{M^*}^{\pi^*}$, $V_{\hat{M}}^{\pi^*}$ the following bound holds $\left\| V_{M^*}^{\pi^*} - V_{\hat{M}}^{\pi^*} \right\|_\infty \leq \frac{\gamma \epsilon}{(1-\gamma)^2}$*

where $\max_{s,a} \|\hat{p}(.|s,a) - p^*(.|s,a)\| \leq \epsilon$ and $\hat{p}$ and $p^*$ are transition of MDP $\hat{M}, M^*$ respectively.

The proof of this lemma is based on the simulation lemma (see Appendix-C). Similar results for RL with imperfect models were reported by (Jiang, 2018).

## 5   POLICY TRANSFER IN SIMULATED ROBOTIC LOCOMOTION TASKS

To evaluate Adapt-to-Learn Policy Transfer in reinforcement learning, we design our experiments using sets of tasks based on the continuous control environments in MuJoCo simulator (Todorov et al., 2012). Our experimental results demonstrate that ATL can adapt to significant changes in transition dynamics. Therefore, we perturb the parameters of the simulated target models for the policy transfer experiments (see Table-1 for original and perturbed parameters of the target mode). To create a challenging training environment, we changed the parameters of the model such that the optimal source policy alone without any learning cannot produce any stable results (see source policy performance in Figure-1). We compare our results against two baselines: (a) Initialized Reinforcement learning (initialized PPO) (Jumpstart-RL (Wang & Mahadevan, 2009)) (b) standalone reinforcement policy learning (PPO) (Schulman et al., 2017).

We experiment with the ATL algorithm on Hopper, Walker2d, and HalfCheetah Environments. The states of the robots are their generalized positions and velocities, and the actions are joint torques. High dimensionality, non-smooth dynamics due to contacts and being under-actuated systems make these tasks very challenging. We use deep neural networks to represent the source and target policy, the details of which are in the Table-2 Appendix-D. The following models are included in our evaluation:

***Slippery Hopper***: is defined through 11-dimensional state space and 3-dimension action space, with reward function defined as $r(t) = (s_{t+1} - s_t)/dt - 10^{-3}\|a\|_2$, and a bonus of $+1$ for being in a non-terminal state. The simulation is terminated upon reaching 1000 steps or hopper toppling. The target model differs in the floor friction and foot joint damping.

***Slippery Fat-Walker2d***: is defined through 17-dimensional state space and 6-dimension action space, with reward function and termination condition defined same as Hopper. The target model differs in the model density and floor friction.

***Fat HalfCheetah***: is defined through 17-dimensional state space and 6-dimension action space, with reward function defined as $r(t) = (s_{t+1} - s_t)/dt - 0.1\|a\|_2$. The simulation is terminated upon reaching 1000 steps. The target model differs in the floor friction coefficient, gravity, and mass.

To establish a standard baseline, we also included the classic cart-pole and Inverted pendulum balancing tasks, based on the formulation (Barto et al., 1983). We also demonstrate the cross-domain transfer capabilities using a model-based variant of the proposed algorithm. The results of policy transfer for Cart-Pole to Inverted Pendulum and Inverted Pendulum to Bicycle transfers are provided in the Appendix-D. Learning curves showing the total reward averaged across three runs of each algorithm are provided in Figure-1. Adapt-to Learn policy transfer solved all the three tasks, yielding quicker learning compared to other baseline methods. These results provide empirical evidence of our hypothesis. Using trajectory KL divergence as intrinsic adaptation reward to adapt source policy to the target, we achieve a more robust and sample efficient policy transfer between two tasks,

| Env | Property | source | Target | %Change |
|---|---|---|---|---|
| Hopper | Floor Friction | 1.0 | 2.0 | +100% |
| HalfCheetah | gravity | -9.81 | -15 | +52% |
| | Total Mass | 14 | 35 | +150% |
| | Back-Foot Damping | 3.0 | 1.5 | -100% |
| | Floor Friction | 0.4 | 0.1 | -75% |
| Walker2d | Density | 1000 | 1500 | +50% |
| | Right-Foot Friction | 0.9 | 0.45 | -50% |
| | Left-Foot Friction | 1.9 | 1.0 | -47.37% |

Table 1: Transition Model and environment properties for Source and Target task and % change

compared to using a warm-start or standalone RL method. Note that the target domain perturbations introduced are significant enough such that source policy alone without any adaptation in the target domain produced no meaningful results (Figure-1). This notion of adaptation in the face of uncertainty is a key advancement over traditional policy transfer, meta-learning, or adversarial RL methods aiming to improve performance by learning a policy over a set of lightly perturbed tasks.

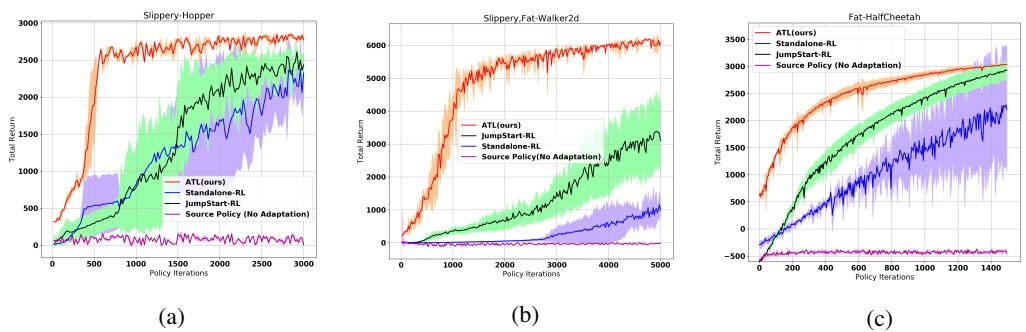

Figure 1: Learning curves for locomotion tasks, averaged across three runs of each algorithm with random Initialization for RL, warm initialization using source policy for ATL and Jumpstart(Warm-Start) methods and Source policy performance in Target Task without any adaptation.

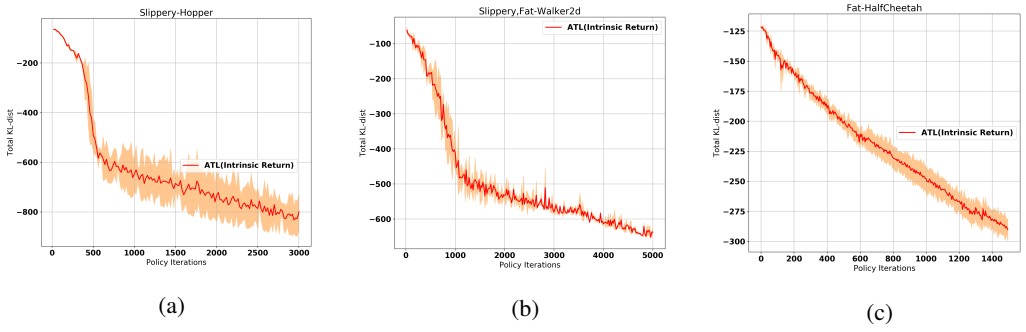

Figure 2: Trajectory KL divergence Total Intrinsic Return $\left( -\sum e^{\zeta_t} \right)$ averaged across three runs.

## 6 CONCLUSION

We introduced a new transfer learning technique for RL: Adapt-to-Learn, that utilizes adaptation of the source policy to target tasks. We demonstrated on nonlinear and continuous robotic locomotion tasks that learning to adapt source policy to the target domain leads to a significant reduction in sample complexity over the prevalent jump-start based approaches. We further also proved theoretical guarantees on the reduced sample complexity of our proposed architecture. There are many exciting directions for future work. A network of policies that can generalize across multiple tasks could be learned based on each new adapted policies. How to train this end-to-end is an important question for meta-learning. The ability of Adapt-to-Learn to handle significant perturbations to the

transition model indicates that it should naturally extend to sim-to-real transfer. Indeed we argue that such adaptation is necessary for real-world robotics, as has been established previously in classical domains like flight control. Another exciting direction is to extend the work to other combinatorial domains (e.g., multiplayer games). We expect, therefore follow on work will find other exciting ways of exploiting such adaptation in RL and machine learning.

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

## A  TOTAL RETURN GRADIENT WITH RESPECT TO POLICY PARAMETERS

The total return which we aim to maximize in adapting the source policy to target is the mixture of environmental rewards and Intrinsic KL divergence reward as follows,

$$\eta_{KL}(\pi_\theta, \pi^*) = \mathbb{E}_{s_t, a_t \sim \tau} \left( p_{\pi_\theta}(\tau) \sum_{t=0}^{\mathcal{H}} r'_t \right) \tag{16}$$

he adaptive policy update methods work by computing an estimator of the gradient of the return and plugging it into a stochastic gradient ascent algorithm.

$$\hat{\pi}_\theta^{*T} = arg \max_{\pi_\theta \in \Pi} P_{Z^n}(\eta_{KL}) \tag{17}$$

$$\theta = \theta + \alpha \hat{g}$$

where $\alpha$ is the learning rate and $\hat{g}$ is the empirical estimate of the gradient of the total discounted return $\eta_{KL}$. The derivation of the gradient term is exactly same as for policy gradient methods.

Taking the derivative of the total return term

$$\nabla_\theta(\eta_{KL}) = \mathbb{E}_{s_t, a_t \sim \tau} \left( \nabla_\theta p_{\pi_\theta}(\tau) \sum_{t=0}^{\mathcal{H}} r'_t \right).$$

Consider the term $\nabla_\theta p_{\pi_\theta}(\tau)$ and evaluate the derivative. Lets consider the term $\log p_{\pi_\theta}(\tau)$. Using the definition of $p_{\pi_\theta}(\tau)$ and we can write the log term as

$$\log p_{\pi_\theta}(\tau) = \log(\rho_0 \pi_\theta(a_0|s_0) p^T(s_1|s_0, a_0) \ldots)$$

$$\log p_{\pi_\theta}(\tau) = \log(\rho_0) + \sum_{t=0}^{\mathcal{H}} \log \pi_\theta(a_t|s_t) + \sum_{t=0}^{\mathcal{H}} p^T(s_{t+1}|s_t, a_t).$$

Taking the derivative wrto $\theta$ we can write,

$$\nabla_\theta \log p_{\pi_\theta}(\tau) = \sum_{t=0}^{\mathcal{H}} \nabla_\theta \log \pi_\theta(a_t|s_t)$$

$$\frac{\nabla_\theta p_{\pi_\theta}(\tau)}{p_{\pi_\theta}(\tau)} = \sum_{t=0}^{\mathcal{H}} \nabla_\theta \log \pi_\theta(a_t|s_t).$$

Therefore we can write

$$\nabla_\theta p_{\pi_\theta}(\tau) = p_{\pi_\theta}(\tau) \sum_{t=0}^{\mathcal{H}} \nabla_\theta \log \pi_\theta(a_t|s_t)$$

Substituting the above expression we can write the total gradient as,

$$\nabla_\theta(\eta_{KL}) = \mathbb{E}_{p_{\pi_\theta}} \left( \sum_{t=0}^{\mathcal{H}} \nabla_\theta \log \pi_\theta(a_t|s_t) \sum_{t=0}^{\mathcal{H}} r'_t \right).$$

## B  PROOF OF THEOREM-4.1

**Theorem B.1** *If the induced class $\mathcal{L}_\Pi$ has uniform convergence in empirical mean property then empirical risk minimization is PAC.*

For notation simplicity we drop the superscript $T$ (for Target domain) and subscript $\theta$ (policy parameters) in further analysis. Unless stated we will using following simplifications $\hat{\pi}^* = \hat{\pi}_\theta^{*T}$ and $\pi^* = \pi_\theta^{*T}$.

**Proof** Fix $\epsilon, \delta > 0$ we will show that for sufficiently large $n \geq n(\epsilon, \delta)$

$$P^n(P(\eta_{KL,\hat{\pi}^*}) - P(\eta_{KL,\pi^*}) \geq \epsilon) \leq \delta \tag{18}$$

Let $\pi^* \in \Pi$ be the minimizer of true return $P(\eta_{KL})$, further adding and subtracting the terms $P_{Z^n}(\eta_{KL,\hat{\pi}^*})$ and $P_{Z^n}(\eta_{KL,\pi^*})$ we can write

$$P(\eta_{KL,\hat{\pi}^*}) - P(\eta_{KL,\pi^*}) =$$
$$P(\eta_{KL,\hat{\pi}^*}) - P_{Z^n}(\eta_{KL,\hat{\pi}^*}) + P_{Z^n}(\eta_{KL,\hat{\pi}^*}) - P_{Z^n}(\eta_{KL,\pi^*}) + P_{Z^n}(\eta_{KL,\pi^*}) - P(\eta_{KL,\pi^*})$$
$$\tag{19}$$

We can divide the above expression in three terms

1. $P(\eta_{KL,\hat{\pi}^*}) - P_{Z^n}(\eta_{KL,\hat{\pi}^*})$

2. $P_{Z^n}(\eta_{KL,\hat{\pi}^*}) - P_{Z^n}(\eta_{KL,\pi^*})$

3. $P_{Z^n}(\eta_{KL,\pi^*}) - P(\eta_{KL,\pi^*})$

Lets consider the term $P_{Z^n}(\eta_{KL,\hat{\pi}^*}) - P_{Z^n}(\eta_{KL,\pi^*})$ in the above expression is always negative semi-definite, since $\hat{\pi}^*$ is a maximizer wrto $P_{Z^n}(\eta_{KL})$, hence $P_{Z^n}(\eta_{KL,\hat{\pi}^*}) \leq P_{Z^n}(\eta_{KL,\pi^*})$ always, i.e

$$P_{Z^n}(\eta_{KL,\hat{\pi}^*}) - P_{Z^n}(\eta_{KL,\pi^*}) \leq 0$$

Next the 1st term can be bounded as

$$P(\eta_{KL,\hat{\pi}^*}) - P_{Z^n}(\eta_{KL,\hat{\pi}^*}) \leq \sup_{\pi \in \Pi}[P_{Z^n}(\eta_{KL}) - P(\eta_{KL})] \leq \sup_{\pi \in \Pi} \|P_{Z^n}(\eta_{KL}) - P(\eta_{KL})\|$$

Similarly upper bound can be written for the 3rd term Therefore we can upper bound the above expression as

$$P(\eta_{KL,\hat{\pi}^*}) - P(\eta_{KL,\pi^*}) \leq 2 \sup_{\pi \in \Pi} \|P_{Z^n}(\eta_{KL}) - P(\eta_{KL})\| \tag{20}$$

From Equation-(18) we have

$$\sup_{\pi \in \Pi} \|P_{Z^n}(\eta_{KL}) - P(\eta_{KL})\| \geq \epsilon/2 \tag{21}$$

Using McDiarmids inequality and union bound, we can state the probability of this event as

$$P^n(\|P_{Z^n}(\eta_{KL}) - P(\eta_{KL})\| \geq \epsilon/2) \leq 2|\Pi|e^{-\frac{n\epsilon^2}{2C^2\mathcal{H}^2}} \tag{22}$$

where $C$ is bound on state space $\mathcal{S}$ and $\mathcal{H}$ is horizon length. The finite difference bound $C$ is obtained assuming transition $p^T, p^S$ follow a Gaussian distributions and the i.i.d trajectories collected are of maximum $\mathcal{H}$ length.

Equating the RHS of the expression to $\delta$ and solving for $n$ we get

$$n(\epsilon, \delta) \geq \frac{2\mathcal{H}^2 C^2}{\epsilon^2} \log\left(\frac{2|\Pi|}{\delta}\right) \tag{23}$$

for $n \geq n(\epsilon, \delta)$ the probability of receiving a bad sample is less than $\delta$. The total number of observed transition is of order $\mathcal{O}(n\mathcal{H})$. Refer supplementary document for details of the proof.

## C  PROOF OF LEMMA-4.3

**Lemma C.1** *Given* $M^*$, $\hat{M}$ *and value function* $V_{M^*}^{\pi^*}$, $V_{\hat{M}}^{\pi^*}$ *the following bound holds* $\left\|V_{M^*}^{\pi^*} - V_{\hat{M}}^{\pi^*}\right\|_{\infty} \leq \frac{\gamma\epsilon}{(1-\gamma)^2}$.

Here $\max_{s,a} \|\hat{p}(.|s,a) - p^*(.|s,a)\| \leq \epsilon$ and $\hat{p}$ and $p^*$ are transition of MDP $\hat{M}, M^*$ respectively.

Note we are deviating from finite horizon assumption and using $\gamma$. To be consistent with previous section we can assume the horizon length $\mathcal{H} = \log_\gamma\left(\frac{\epsilon_v}{2V_{max}}\right)$. Where $\epsilon_v$ is truncation error in total infinite return.

**Proof** For any $s \in \mathcal{S}$

$$|V_{\hat{M}}^{\pi^*}(s) - V_{M^*}^{\pi^*}(s)|_\infty$$
$$= |r(s,a) + \gamma \left\langle \hat{p}(s'|s,a), V_{\hat{M}}^{\pi^*}(s') \right\rangle - r(s,a) - \gamma \left\langle p^*(s'|s,a), V_{M^*}^{\pi^*}(s') \right\rangle |_\infty$$

Add and subtract the term $\gamma \left\langle p^*(s'|s,a), V_{\hat{M}}^{\pi^*}(s') \right\rangle$

$$= |\gamma \left\langle \hat{p}(s'|s,a), V_{\hat{M}}^{\pi^*}(s') \right\rangle - \gamma \left\langle p^*(s'|s,a), V_{\hat{M}}^{\pi^*}(s') \right\rangle + \gamma \left\langle p^*(s'|s,a), V_{\hat{M}}^{\pi^*}(s') \right\rangle - \gamma \left\langle p^*(s'|s,a), V_{M^*}^{\pi^*}(s') \right\rangle |_\infty$$
$$\leq \gamma |\left\langle \hat{p}(s'|s,a), V_{\hat{M}}^{\pi^*}(s') \right\rangle - \left\langle p^*(s'|s,a), V_{\hat{M}}^{\pi^*}(s') \right\rangle| + \gamma |\left\langle p^*(s'|s,a), V_{\hat{M}}^{\pi^*}(s') \right\rangle - \gamma \left\langle p^*(s'|s,a), V_{M^*}^{\pi^*}(s') \right\rangle |_\infty$$
$$\leq \gamma |\hat{p}(s'|s,a) - p^*(s'|s,a)|_\infty |V_{\hat{M}}^{\pi^*}(s')|_\infty + \gamma |V_{\hat{M}}^{\pi^*}(s) - V_{M^*}^{\pi^*}(s)|_\infty$$

Using the definition of $\epsilon$ in above expression, we can write

$$|V_{\hat{M}}^{\pi^*}(s) - V_{M^*}^{\pi^*}(s)|_\infty \leq \gamma \epsilon |V_{\hat{M}}^{\pi^*}(s')|_\infty + \gamma |V_{\hat{M}}^{\pi^*}(s) - V_{M^*}^{\pi^*}(s)|_\infty$$

Therefore

$$|V_{\hat{M}}^{\pi^*}(s) - V_{M^*}^{\pi^*}(s)|_\infty \leq \frac{\gamma \epsilon |V_{\hat{M}}^{\pi^*}(s')|_\infty}{1-\gamma}$$

Now we solve for expression $|V_{\hat{M}}^{\pi^*}(s')|_\infty$. We know that this term is bounded as

$$|V_{\hat{M}}^{\pi^*}(s')|_\infty \leq \frac{R_{max}}{1-\gamma}$$

where $R_{max} = 1$, therefore we can write the complete expression as

$$|V_{\hat{M}}^{\pi^*}(s) - V_{M^*}^{\pi^*}(s)|_\infty \leq \frac{\gamma \epsilon}{(1-\gamma)^2}$$

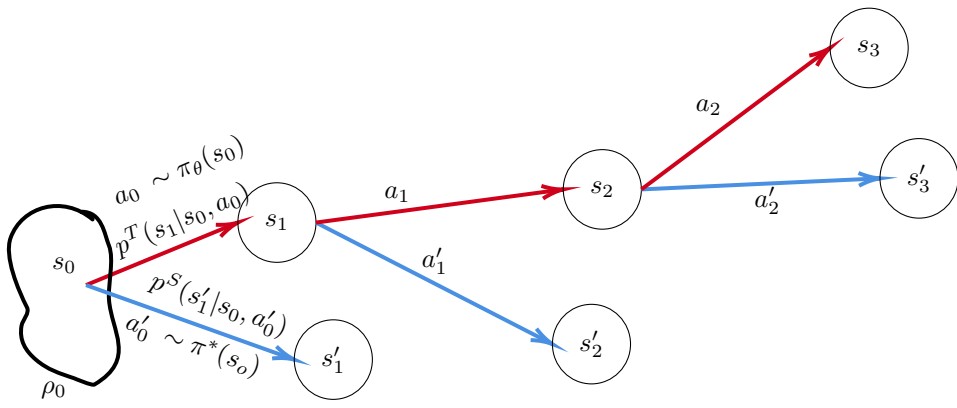

Figure 3: Target Trajectory under policy $\pi_\theta$ and local trajectory deviation produced by source optimal policy $\pi^*$ and source transition $p^S$

# D    ADDITIONAL RESULTS AND DETAILS OF EXPERIMENTAL DOMAIN

To establish a standard baseline, we also include the results for classic cart-pole and Inverted pendulum balancing tasks, based on the formulation Barto et al. (1983). We also demonstrate the cross-domain transfer capabilities using a variant of the proposed algorithm Joshi & Chowdhary (2018) and the results for Cart-Pole to Inverted Pendulum and Inverted Pendulum to Bicycle transfers are provided in the following sections.

|  | Hopper | Walker2d | HalfCheetah |
|---|---|---|---|
| State Space | 12 | 18 | 17 |
| Control Space | 3 | 6 | 6 |
| Number of layers | 3 | 3 | 3 |
| Layer Activations | tanh | tanh | tanh |
| Total num. of network params | 10530 | 28320 | 26250 |
| Discount | 0.995 | 0.995 | 0.995 |
| Learning rate | $1.5 \times 10^{-5}$ | $8.7 \times 10^{-6}$ | $9 \times 10^{-6}$ |
| $\beta$ init | 1.0 | 1.0 | 1.0 |
| $\beta$-anneal coeff | 0.998 | 0.998 | 0.998 |
| Batch size | 20 | 20 | 5 |
| Policy Iter | 3000 | 5000 | 1500 |

Table 2: Policy Network details and Network learning parameter details

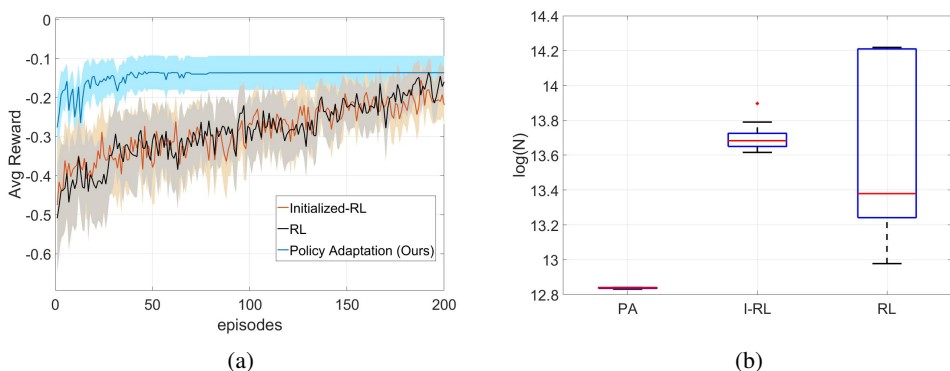

(a)                                          (b)

Figure 4: Policy Transfer from Inverted Pendulum to Non-stationary Inverted pendulum: (a) Average Rewards and (b) Training length, TA-TL(ATL ours), UMA-TL(Jumpstart-RL) and Stand-alone RL

## D.1 INVERTED PENDULUM (IP) TO TIME-VARYING IP

We demonstrate our approach for a continuous state domain, Inverted Pendulum (IP) swing-up and balance Figure-4. The source task is the conventional IP domain. The target task differs from the source task in the transition model. The target task is a non-stationary inverted pendulum, where the length and mass of the pendulum are continuously time varying with function $L_i = L_0 + 0.5cos(\frac{\pi i}{50})$ and $M_i = M_0 + 0.5cos(\frac{\pi i}{50})$, where $L_0 = 1$, $M_0 = 1$ and $i = 1 \ldots N$. The state variables describing the system are angle and angular velocity $\{\theta, \dot{\theta}\} \in [-\pi, \pi]$. The RL objective is to swing-up and balance the pendulum upright such that $\theta = 0, \dot{\theta} = 0$. The reward function is selected as $R(\theta, \dot{\theta}) = -10|\theta|^2 - 5|\dot{\theta}|^2$, which yields maximum value at upright position and minimum at the down-most position. The continuous action space is bounded by $T \in [-1, 1]$. Note that the domain is tricky, since full throttle action is assumed to not generate enough torque to be able to swing the pendulum to the upright position, hence, the agent must learn to swing the pendulum back and forth and leverages angular momentum to go to the upright position.

## D.2 ROBUSTNESS TO NEGATIVE TRANSFER

We demonstrate that the proposed transfer is robust to negative transfers, that is, cases in transfer learning where naive initialization of the target using source policy could be detrimental. We demonstrate this through an inverted pendulum upright balance task with the sign of control flipped. That is, we use an inverted pendulum model as both source and target systems, but the target is the inverse of the source model with the sign of the control action flipped. It should be noted that dealing with a change of sign in control has been considered a highly challenging problem in adaptive control Chowdhary et al. (2013). We demonstrate that ATL is immune to negative transfers. Figure

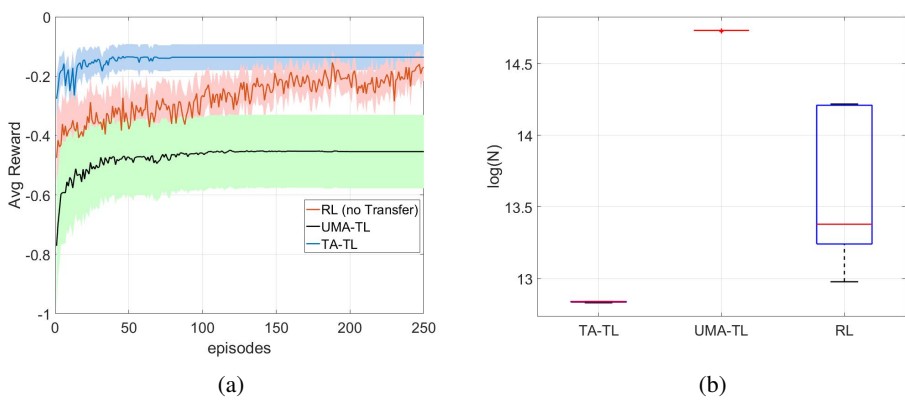

(a)  (b)

Figure 5: Negative Transfer Inverted Pendulum: (a) Average Reward and (b) Training length

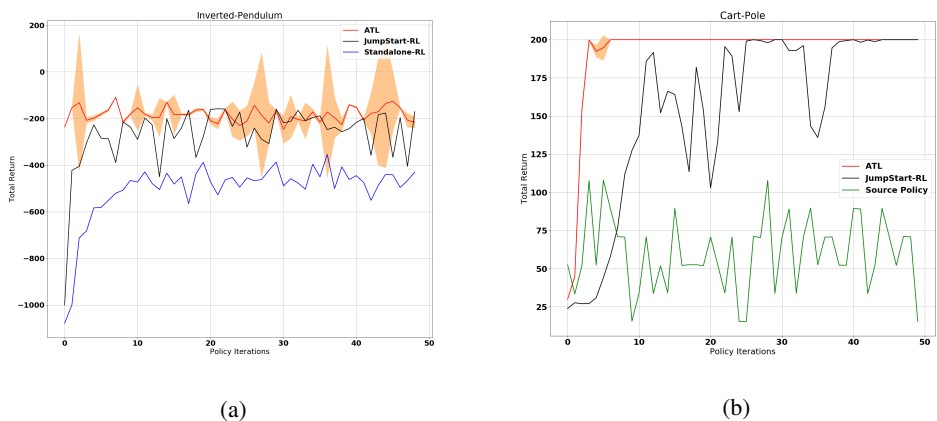

(a)  (b)

Figure 6: Adap-to-Learn Policy Transfer (a) Inverted-Pendulum (b) Cart-Pole

5a and 5b demonstrate convergence to average maximum reward with lesser training samples for proposed ATL method compared to Initialized-RL(UMA-TL) and standalone-RL methods. It is to be observed that UMA-TL method suffers from a negative transfer phenomenon. The agent under UMA-TL converges to a much lower average reward by getting stuck in local minima and never achieves the upright balance of the pendulum.

Also, the samples observed by UMA-TL in learning the task is much higher compared to no transfer (RL) and proposed ATL methods. If the source and target are not sufficiently related or the features of source task do not correspond to the target, the transfer may not improve or even decrease the performance in target task leading to negative transfer. We show that the UMA-TL suffers from a negative transfer in these results, whereas the performance of presented ATL is much superior compared to UMA-TL and RL(learning from scratch).

# E    CROSS DOMAIN TRANSFER

## E.1    CART-POLE TO BICYCLE:

Bicycle balancing is a challenging physical problem, especially when the bicycle velocity is below the critical velocity $V_c = 4m/s$ to $5m/s$. We set the bicycle velocity to be $V < V_c$ i.e., $V = 2.778m/s$ such that the bicycle becomes unstable, and active control is required to maintain stability. The simulation itself is very high fidelity and realistic, which was designed for studying the physics of the bicycle.

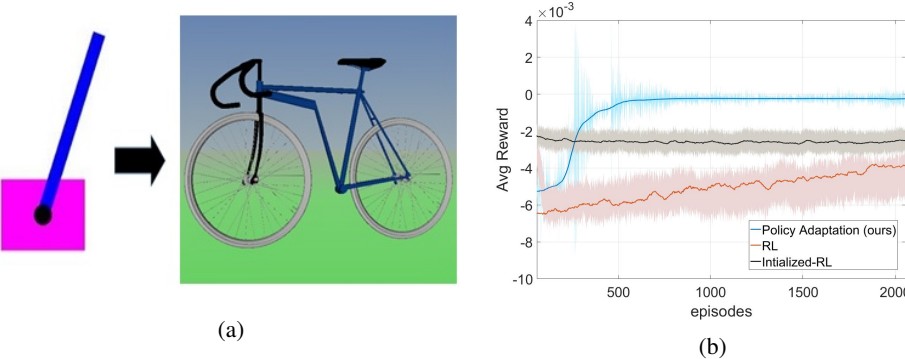

(a)

(b)

Figure 7: (a) Cart-Pole and Bicycle Domain (b)Average Rewards for TA-TL (ours), UMA-TL (jump-start) and RL

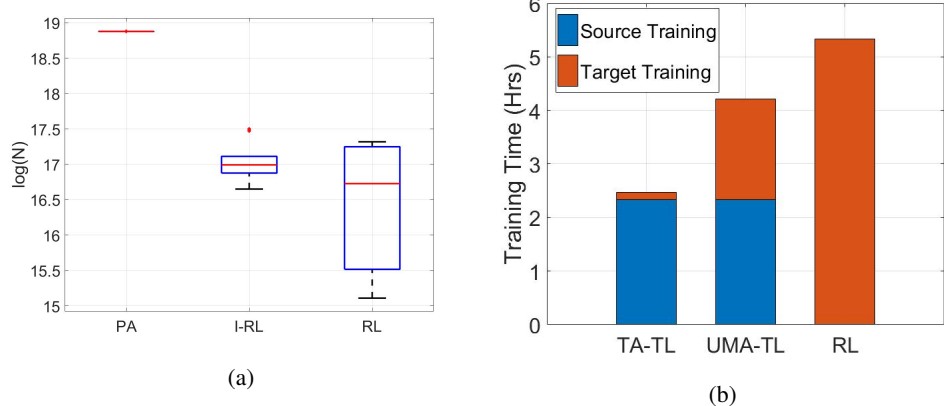

(a)

(b)

Figure 8: Policy Transfer from Cart-Pole to Bike Balancing: (a) Total simulation time (in seconds) the agent was able to balance the bike in training (b) Total time required to solve the task for TA-TL (ours), UMA-TL (jumpstart) and RL

The states of the bicycle task are angle and angular velocity of the handlebar and the bike from vertical $(\theta, \dot{\theta}, \omega, \dot{\omega})$ respectively. For the given state the agent is in, it chooses a continuous action of applying torque to the handlebar, $T \in [-2Nm, 2Nm]$ trying to keep the bike upright. The details of bicycle dynamics are beyond the scope of this paper; interested readers are referred to Randlov & Alstrom (1998); Åström et al. (2005), and references therein.

We use the Cart-Pole as source task for learning to balance a bicycle. The bicycle balance problem in principle is similar to that of cart-pole, since the objective is to keep the unstable system upright. The objective of balance is achieved in both the systems by moving in the direction of fall, which is termed as a non-minimum phase behavior in the controls system literature. The control in the cart pole affects the angle of the pole, by moving the cart such that it is always under the pole. In the bicycle, the control is to move the handlebar in the direction of fall. However, balancing the bike is not straightforward, to turn the bike under itself, one must first steer in the other direction before turning in the direction of fall; this is called counter-steering Åström et al. (2005). We observe that both cart pole and bicycle have this commonality in their dynamical behaviors, as both the system have a non-minimum phase that is the presence of unstable zero. This similarity qualifies the cart-pole system as an appropriate choice of source model for bicycle balance task.

Cart pole is characterized by state vector $[x, \dot{x}, \theta, \dot{\theta}]$, i.e., position, the velocity of cart and angle, angular velocity of the pendulum. The action space is the force applied to the cart $F \in [-1N, 1N]$. For the mapping between the state space of bicycle and cart pole model, we use Unsupervised Manifold Alignment(UMA) to obtain this mapping Wang & Mahadevan (2009). We do not report

the training time to learn the intertask mapping since it is common to both ATL and UMA-TL methods. Figure 7b and 8 shows the quality of transfer for ATL through faster convergence to average maximum reward with lesser training samples compared to UMA-TL and RL methods.

## E.2 Mountain Car (MC) to Inverted Pendulum (IP)

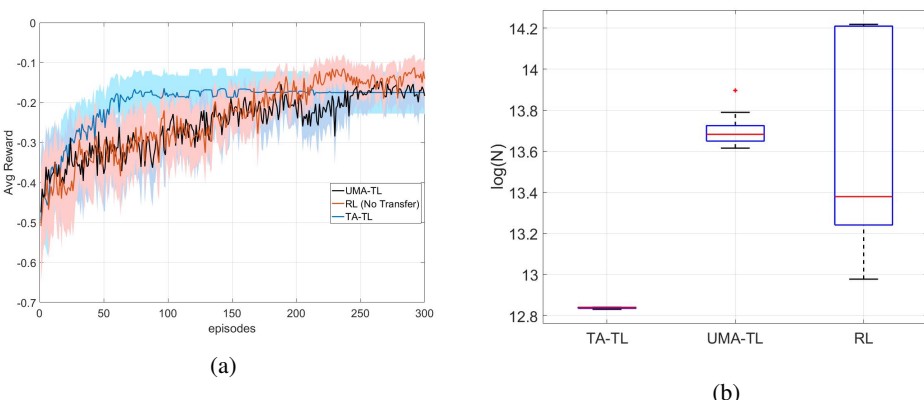

(a)

(b)

Figure 9: Policy Transfer from Mountain Car to Inverted Pendulum: (a) Average Rewards and (b) Training length

We also demonstrate the cross-domain transfer between mountain car to an inverted pendulum. The source and target task are characterized by different state and action space. The source task MC is a benchmark RL problem of driving an underpowered car up a hill. The dynamics of MC are described by two continuous state variables $(x, \dot{x})$ where $x \in [-1.2, 0.6]$ and $\dot{x} \in [-0.07, 0.07]$ and one continuous action $F \in [-1, 1]$. The reward function is proportional to the negative of the squared distance of the car from goal position. The target task is conventional IP with state $(\theta, \dot{\theta}) \in (-\pi, \pi)$ and action $T \in [-1, 1]$. We present the performance of transfer methods based on sample efficiency in learning the target task and speed of convergence to the maximum average reward. Similar to the bicycle domain transfer Figure 9a and 9b shows the quality of transfer for ATL through faster convergence to average maximum reward with lesser training samples compared to UMA-TL and RL methods.

