# OpenReview forum: "Adapt-to-Learn: Policy Transfer in Reinforcement Learning"
_ICLR.cc/2020/Conference — Reject_

### Official Review · AnonReviewer3 · 2019-10-22
**Official Blind Review #3**

**Rating:** 1

**Review:**

Summary
-------
The authors propose an algorithm for transferring policies across domains differing in their transition model. The idea is to "regularize" the target policy being learned to generate a trajectory distribution similar to the one induced by the source optimal policy in the respective MDP. The method is proved effective in standard simulated robotic tasks (Mujoco).

Major comments
--------------
The problem of transfer in RL is fundamental for scaling RL algorithms to real domains. The presented ideas are interesting and could potentially inspire other works in this field. The paper is well-written and easy to read. However, I believe there might be some technical issues with the derivations that overall make me doubt about the significance of this work. My detailed comments follow.

1. Under Assumption 1, the step from (6) to (7) seems flawed. The definition of the KL divergence requires that p is absolutely continuous w.r.t. q. However, under Ass. 1, whenever the trajectory \tau contains an action that is not taken by \pi^*, q(\tau) = 0, which implies that p(\tau) must be zero as well. The only way to enforce this is to set \pi_\theta = \pi^*, and the solution is trivial. The main issue is that (6) writes the log term of the KL using different action random variables for the numerator and the denominator (a and a'), but these must be the same. Therefore, one cannot simply set to 1 the optimal source actions. Notice also that the action space is defined continuous, hence the value of the pdf in the chosen deterministic action is infinity, not 1.

2. (12) seems to be wrong too. Note that the reward function defined here (r') contains the policy, hence it is a function of \theta and must be differentiated. So the standard REINFORCE estimator written here does not work theoretically.

3. Assumption 2 basically requires that either we can get new samples from the source domain "on-demand", or we have a good simulated model. Both assumptions are often violated in practice. Could the authors elaborate more on why they believe these assumptions to be reasonable?

4. Related to the previous point, assuming the one-step transition models to be Gaussian is convenient, and often reasonable, in practice. But how do we choose the std \sigma? What if the true models are deterministic?

5. I would disagree with the last sentence of section 3.2.1, which states that minimizing the KL divergence is more sample efficient than standard RL/policy-search methods. One can always define an MDP where the immediate reward is the KL term and that is a standard RL problem. Furthermore, if we compute the expected return when the reward is (8), decomposing the log into the sum of two logs, one of the two expectations we obtain is of the form: E_\pi[-(s_{t+1} -s'_{t+1})^2], which is a standard "tracking" problem.

6. I was quite confused by the theoretical section. The introduction states that the proposed method enjoys a sample complexity of O(nH), where I guess n is the number of state-action pairs. Where does that dependence appear in Section 4? Theorem 4.1 is a standard supervised learning bound. How do we read that in RL? What is C? Is |\Pi| the cardinality of the set of policies (assumed finite)? Or is it the pseudo-dimension of the policy space? Lemma 4.3 is also a know result in the literature, but what does it tell us about the proposed approach? The idea is that if I have an accurate model \hat{p} I am also close in value function to the true MDP. But what is \hat{p} here? The proposed algorithm is not computing any explicit transition model.

Minor comments
--------------
- In the last sentence of the introduction, it is not clear what n is in O(nH)
- In the preliminaries, is the reward R: S x A -> R_+ assumed to be positive or is that a typo?
- MDPs are defined with a discount factor \gamma which is however missing in (1) and the next equation (V)
- In the preliminaries, it is not clear whether the rewards differ or not between domains. It is important to specify that they do not.
- In (2), T is defined to map R^n -> R^n. What is n? Is it the number of discrete states? Wasn't the state space continuos?
- I noticed that many sentences in the paper begin without a space after the period of the previous sentence.
- In section 3, the KL divergence (e.g., eq. 3,7) is written as an expectation that contains the pdf p(\tau). Either only the log term is written in the expectation, or E is replaced by an integral.
- In (12), the sum should be up to H and not infinity

**Experience Assessment:**

I have published one or two papers in this area.

**Review Assessment: Checking Correctness Of Derivations And Theory:**

I carefully checked the derivations and theory.

**Review Assessment: Checking Correctness Of Experiments:**

I assessed the sensibility of the experiments.

**Review Assessment: Thoroughness In Paper Reading:**

I read the paper thoroughly.

---

### Official Review · AnonReviewer1 · 2019-10-24
**Official Blind Review #1**

**Rating:** 6

**Review:**


This paper addresses the actively studied problem of efficiently transferring policies across domains in reinforcement learning. Authors propose a framework to transfer policies between tasks in domains with significantly different state transition. The proposed algorithm is based on a policy adaptation mechanism, with the idea that provided that a source optimal policy of a task is available, that policy is adapted to derive the optimal policy of the target task at a low sample complexity.

The paper is well written and the proposed algorithm is novel and original. The algorithm is very clearly described and theoretical bounds are provided. The performance is evaluated experimentally on a set of tasks, showing promising results.

Cons:
- Some important previous  works are missing from the related work section. For example, [1, 2] are also concerned with transferring policy across task that share the same domain.
- The experimental section could benefit from comparing to other transfer RL baselines such as [1].
- Assumption 2 should be discussed; any suggestion on how to model the KL approximation if the assumption does not hold?

Minor comments for clarity:
- some acronyms are introduced without ever being spelled out: D-RL, TL, MDP, PAC
- page 3, equation before (2), is \gamma = 1?
- the use of s' for the optimal state, a' for optimal action and then r' for total reward is quite confusing. Specially because s' is in general used to describe a subsequent state in the literature.
- page 7, first line 'discount' instead of 'discout'
- font on figures 1 and 2 is very small

1. Barreto et al, Transfer in Deep Reinforcement Learning Using Successor Features and Generalised Policy Improvement, 2018
2. Ma, Wen, and Bengio Universal Successor Representations for Transfer Reinforcement Learning, 2018

**Experience Assessment:**

I have published one or two papers in this area.

**Review Assessment: Checking Correctness Of Derivations And Theory:**

I did not assess the derivations or theory.

**Review Assessment: Checking Correctness Of Experiments:**

I assessed the sensibility of the experiments.

**Review Assessment: Thoroughness In Paper Reading:**

I read the paper at least twice and used my best judgement in assessing the paper.

---

### Official Review · AnonReviewer2 · 2019-10-26
**Official Blind Review #2**

**Rating:** 6

**Review:**

Summary
This paper tackles the problem of transferring a policy from source to target MDP, which differ in the state transition function. The idea is to add an additional cost that is the KL divergence between the trajectory likelihood under target policy (being learned) and target dynamics and the trajectory likelihood under the source policy (assumed optimal and deterministic) and source dynamics. The intuition is that the target policy will learn to match the state distribution of the optimal source policy. Results on MuJoCo locomotion robots with varying physics show that the proposed method performs better on target than warm-started RL or learning from scratch.

I think the problem of transferring knowledge from one task to another in RL is very important for RL to be applicable to more real-world scenarios.

Concerns / Questions
Line 7 of Alg1 is confusing because it refers to a “target task model”, but in Assumption 2, it says only a model of the source transition function is needed. I think it makes sense that only the source transition model is needed because the target next state is given by experience.
I think the combined assumptions of a) access to expert behavior (same as DAGGER) and b) that the MDPs differ only in dynamics functions and c) access to the source transition model are rather strong. I think (b) is a special case of transfer learning - a lot of transfer learning is concerned with changing reward functions as well, which this method wouldn’t apply to. I think this could be made more clear in the paper. It would be good if all these assumptions were made clear and discussed.
I think the related work section is missing important areas of research in imitation learning and meta-reinforcement learning. For imitation learning, the approach strikes me as bearing similarity to PRECOG: PREdiction Conditioned On Goals in Visual Multi-Agent Settings (Rhinehart et al.), and the topic of imitation learning should be discussed in general. For meta-RL, mentioning that it shares the same goal of transfer and citing a few main works (e.g. Duan et al. 2016, Wang et al. 2016, Finn et al. 2017 etc) would be good.

Writing Suggestions
Some terms used throughout the paper are quite unclear (e.g., “unsupervised RL”, “intrinsic adaptation reward”, “supervised reference trajectory tracking”). I suggest standardizing and defining terms early to avoid unnecessary confusion.
Writing the Bellman operator and the value function equations in Section 2 don’t seem very relevant as they are I think never used again?
Sections 3.1 and 3.2 are quite difficult to understand on first read (e.g., what does “point-wise local trajectories” mean?).
I find Section 3.2.1 a bit misleading, “The optimization is more akin to supervised learning” - I agree the KL minimization is essentially imitation learning, but you are still doing policy search in addition to it?


**Experience Assessment:**

I have published one or two papers in this area.

**Review Assessment: Checking Correctness Of Derivations And Theory:**

I assessed the sensibility of the derivations and theory.

**Review Assessment: Checking Correctness Of Experiments:**

I carefully checked the experiments.

**Review Assessment: Thoroughness In Paper Reading:**

I read the paper thoroughly.

---

### Decision · Program_Chairs · 2019-12-19

**Decision:**

Reject

**Comment:**

This paper considers inter-domain policy transfer in reinforcement learning. The proposed approach involves adapting existing policies from a source task to a target task by adding a cost related to the difference between the dynamics and trajectory likelihoods of the two tasks.

There are three major problems with this paper as it stands, as pointed out by the reviewers. Firstly, the "KL divergence" is not a real KL divergence and seems to be only empirically motivated. Then, there are issues with the derivative of the policy gradient. Finally, the theory is not well connected to the proposed algorithm. The rebuttals not only failed to convince the reviewer that raised these issues, but another reviewer lowered their score as a result of these raised points.

This is a really interesting idea with compelling experiments, but must be rejected at this point for the aforementioned reasons.